# Optimal restoration for pollination services increases forest cover while doubling agricultural profits

Sofía López-Cubillos [1,2]*, Eve McDonald-Madden[1], Margaret M. Mayfield[3], Rebecca K. Runting[2]

**1** School of Earth and Environmental Science and Centre for Biodiversity and Conservation Science, University of Queensland, St Lucia, Brisbane, Queensland, Australia, **2** School of Geography, Earth and Atmospheric Sciences, University of Melbourne, Parkville, Melbourne, Victoria, Australia, **3** School of BioSciences, University of Melbourne, Parkville, Melbourne, Victoria, Australia

* sofia.lopezcubillos@unimelb.edu.au

**Data Availability Statement:** All relevant data are within the paper and its Supporting Information files.

## Abstract

Pollinators are currently facing dramatic declines in abundance and richness across the globe. This can have profound impacts on agriculture, as 75% of globally common food crops benefit from pollination services. As many native bee species require natural areas for nesting, restoration efforts within croplands may be beneficial to support pollinators and enhance agricultural yields. Yet, restoration can be challenging to implement due to large upfront costs and the removal of land from production. Designing sustainable landscapes will require planning approaches that include the complex spatiotemporal dynamics of pollination services flowing from (restored) vegetation into crops. We present a novel planning framework to determine the best spatial arrangement for restoration in agricultural landscapes while accounting for yield improvements over 40 years following restoration. We explored a range of production and conservation goals using a coffee production landscape in Costa Rica as a case study. Our results show that strategic restoration can increase forest cover by approximately 20% while doubling collective landholder profits over 40 years, even when accounting for land taken out of production. We show that restoration can provide immense economic benefits in the long run, which may be pivotal to motivating local landholders to undertake conservation endeavours in pollinator-dependent croplands.

## Introduction

To address the global biodiversity crisis, landscape restoration has gained increasing traction [1], leading to initiatives such as the "Bonn challenge," which aims to restore 350 million hectares of land [2], and the United Nations' declare 2020–2030 as the "Decade on Ecosystem Restoration." At the same time, food demand is expected to increase by 50% to 70% by 2050 [3], potentially expanding agricultural land use and threatening biodiversity [4] along with crucial ecosystem services [5]. Therefore, restoration can be perceived as both as a competing activity with agricultural land use and crop production [6,7] and as an action than can cost effectively protect biodiversity and increase agricultural outcomes through the flow of ecosystem services critical to agriculture, such as pollination [8]. Pollination is a significantly important service in

**Funding:** SLC was supported by a Colombian Ministry of Education grant (COLCIENCIAS, No. 728) and the Research Training program provided by the Graduate School from the University of Queensland. RKR is funded by an Australian Research Council DECRA Fellowship (DE210100492). EMM is supported by an Australian Research Council Future Fellowship. The funders had no role in study design, data collection and analysis, decision to publish, or preparation of the manuscript.

**Competing interests:** The authors have declared that no competing interests exist.

agricultural landscapes as 75% of agricultural crops benefit from it [9]. Studies have quantified that the restoration of native flora adjacent to some berry crops [10] resulted in yield increases of 5% to 6% [11,12] and by 1.5 kg per tree for all mango cultivars growing near wild flowering areas [13]. However, whether these increased yields can compensate for the costs of setting aside restored lands at a landscape scale remains unclear.

In addition, not all biodiversity nor the ecosystem services it provides is realised immediately after initial restoration activities, a reality that represents a hurdle to the widespread uptake of restoration measures in agricultural landscapes [14]. For example, restoration efforts near tea, shade coffee, cardamom, and eucalyptus plantations took 9 to 17 years to see increased bird abundance in active restoration sites [15]. Further, restoration incurs upfront and ongoing financial costs can be substantial to ensure restoration success [16]. Therefore, understanding how restoration provides financial benefits over time via ecosystem services to agriculture is also essential. A good example is the increased profitability after 3 to 5 years following restoration near blueberry fields due to the benefits from pollination services [11]. As pollination is a key service to many crops, it is therefore important to quantify the bee abundance and the economic benefits from different restored patches with a range of restoration ages across the landscape. Ultimately, the question remains, where and how much land can be restored for biodiversity conservation over time, without resulting in declines in agricultural profits?

Spatial optimization is a modelling approach that allows a quantitative analysis of the trade-offs between conservation and agriculture production using ecosystem services [17]. Furthermore, Kennedy and colleagues [18] and López-Cubillos and colleagues [19] have found that explicitly accounting for ecosystem services in spatial optimization under different scenarios reveals positive net social and environmental benefits by increasing income or social equity. Previous studies have explored the economic importance of restored lands to crop pollination [11], including using spatial optimization tools [20–22]. However, no studies, to our knowledge, have simultaneously considered demand for pollination services at a landscape scale (rather than at a farm scale as done by Blaauw and Isaacs [11]) and the changes for pollinator abundance with the services they provide in a dynamic temporal manner over a specific timeframe, including the economic returns they deliver.

We used coffee as a case study given its importance as a cash crop in the Global South [23] and because it is known to benefit from pollination provided by wild bees [24–27]. Indeed, while coffee can self-pollinate, the yields of *Coffea arabica* in particular can increase from 10% to 30% due to bee pollination [25,26]. The specific system we used was coffee in Tarrazú, one of the main coffee producing canton (administrative division) in Costa Rica. Coffee production in this region has been shown to decrease by 18% when crops are too far from forest [28,29]. Using this system, we answer the following questions: (1) To what extent can the strategic allocation of land for restoration achieve both conservation and crop production goals? (2) How do these benefits change over time? (3) How important is agricultural expansion to these outcomes?

## Methods

In this paper, we present a novel optimization framework for the spatial and temporal allocation of restoration in agricultural landscapes (Fig 1). The framework aims to maximise coffee profitability and forest restoration and retention in 2 different agricultural contexts. In the first context, there is no intention of expanding coffee production within the remnant forest but instead it only seeks to restore agricultural land (named as "Only restoration" here after). In the second context, both restoration and agricultural expansion are explored (named as "Expansion and restoration" here after). We also used a "Baseline" context, which represents the current landscape without any coffee expansion nor restoration in agricultural lands. To

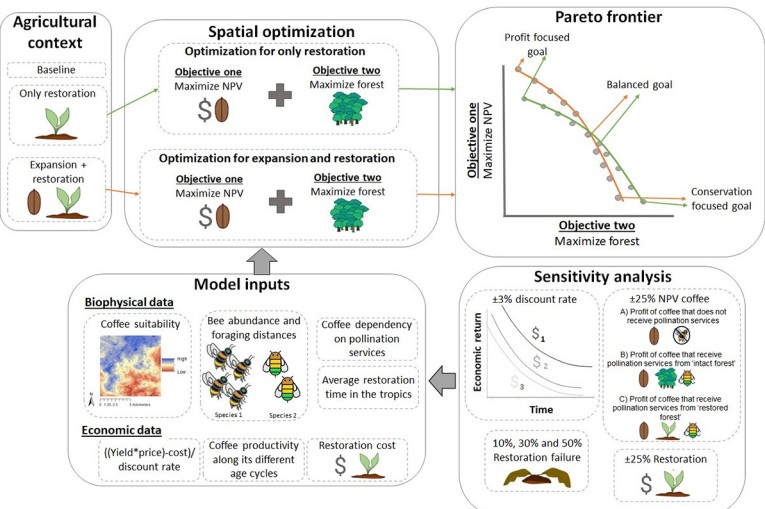

**Fig 1. Spatial optimization framework.** In the "Agricultural context" box, 3 different contexts are represented. The "Baseline" context is the current landscape where there is no coffee expansion or restoration. In the "Only restoration" context, restoration is allowed to happen within the coffee cropland, but no coffee expansion is allowed. In the "Expansion and restoration" context, both coffee expansion in intact forest and coffee cropland restoration is allowed across the landscape. The "Model input" box represents all the biophysical and economic data needed to run the optimization (see Methods). In the "Optimization" boxes, we want to simultaneously maximise the coffee NPV and forest habitat for each agricultural context. In the "Sensitivity analysis" box, 4 sets of sensitivity analyses are conducted on key variables (see Methods). Finally, the "Pareto frontier" box represents the trade-offs between maximising the NPV or forest. In this box, we can highlight 3 main goals for each optimization: (1) The "Conservation Focus Goal" gives more priority to restoration; (2) the "Profit Focus Goal" gives more priority to agricultural profit (NPV), and (3) the "Balanced Goal" aims to find an equilibrium between both objectives. Most of the clip art were created on the website https://www.autodraw.com/ that is a Creative Commons Attribution 4.0 International License and only a couple (plants for restoration and also restoration failure) were drawn by hand.

achieve this, we accounted for the spatial and temporal variability in the economic benefits from active restoration in full sun coffee, alongside the costs of setting aside land.

Although forest friendly shaded coffee is increasingly common [30], monocultures of full sun coffee still dominate coffee production in much of the world [31,32]. While natural regeneration can be a cost-effective option for biodiversity protection and the maintenance of ecosystem services, this action is usually most possible in lands of marginal production [33]. For example, in countries like Puerto Rico and Brazil, declining rural populations in coffee landscapes is opening the window for more natural regeneration as land is abandoned [34]. However, the aim of this paper is to provide a framework for expansion of natural areas in landscapes used for active agricultural production, so we focused on active restoration, as it has been shown to provide effective outcomes in coffee regions in Costa Rica. For example, Holl and colleagues [35] show that active nucleation (clusters of plants within a large area) can attract greater numbers of birds, bats, epiphyte species, and tree recruitment compared to natural regeneration in coffee and cattle lands after a decade of restoration. Therefore, to model the costs of active restoration, we estimated the number of plants and materials for a 50 × 50 m plot size using values from Holl and colleagues [36]. The land cover data was obtained from the Technological Institutional Repository of Costa Rica (S1 Appendix—2.1).

## Coffee yield and profit calculation

In this study, we adapted the methodology proposed by López-Cubillos and colleagues [17] for assessing part of the yield calculations and for the optimization analysis. We divided our study

area (Fig 2 baseline) into 62,248 grid cells, each 1,600 m$^2$ (40 × 40 m), with smaller cells at the borders and corners of the study area (i.e., 800 m$^2$ and 400 m$^2$, respectively). Although cells are small, studies have found that pollination services can be enhanced within such patch sizes in other insect-pollinated crops such as mango and almonds [13,37]. Furthermore, because most farmers from these regions are smallholders [38,39], the size of these cells are a realistic scale of

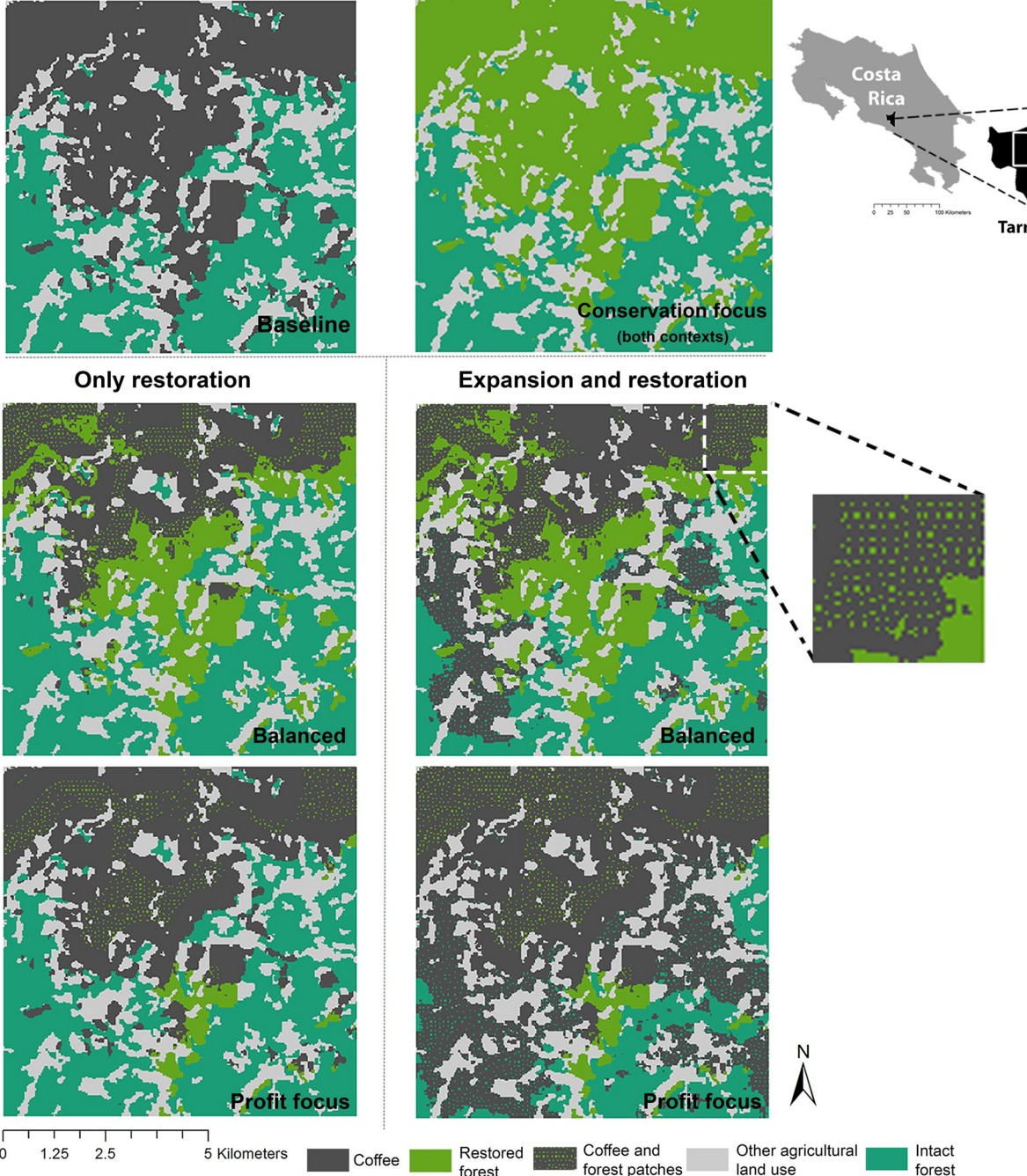

**Fig 2. Baseline and spatial representation of the different goals for each context in the last time step (year 40) as described below in section "Planning goals and optimization analysis."** The data underlying this figure can be found in S1 Data in tabs 1–6 for each map. The base map layer was taken from The World Bank Data Catalog and do not require credit because of public domain (https://datacatalog. worldbank.org/search/dataset/0038272/World-Bank-Official-Boundaries).

what could be restored or protected. For each cell, we identified the current land use type and calculated coffee yield and profitability. We used 3 commonly used parameters in the literature to assess coffee yields: coffee land suitability, coffee plant age, and the potential pollination services [29,40–45]. Other parameters such as farm management practices (the use of fertilises, water, etc.) could also be included in the assessment of coffee yields [46,47]; however, because farm practices can vary widely across the agricultural landscape and are difficult to ascertain, we decided to exclude these factors.

Coffee yields were calculated using a distance decay function proposed by Lonsdorf and colleagues [48] and Ricketts [29] as follows:

$$Y_o = Y_{max}\left(1 - v_c + v_c \frac{P_o}{P_o + k_c}\right) \tag{1}$$

Here, $Y_o$ represents the expected yield, $Y_{max}$ is the maximum yield possible that varies with coffee suitability (map taken from López-Cubillos and colleagues [17], Fig 2.1 in S1 Appendix), $k_c$ is a half-saturation constant, $v_c$ is the amount of coffee dependence on pollinators, and $P_o$ is the scaled bee abundance with respect to the maximum number of observed individuals in a sampling site. Most of the parameters on Eq 1 were taken from Ricketts' [29] and López-Cubillos [17]. $Y_{max}$ changed with the coffee plant age, as coffee yields change across plant maturity, being low at ages 2 to 3 and 15 to 20 years, medium for ages 4 to 6 and 10 to 14 years, and high for plants aged between 7 and 9 years (S1 Appendix—2.2). Once bee abundance and coffee age were incorporated, we multiplied these values by land suitability value (ranging from 1 to 5, with 1 being the lowest and 5 the highest level of suitability) for each grid cell.

The time required to reach restoration benefits may vary according to the objective assessed. For example, restoring crop fields with small flowering patches can start showing benefits within only 10 years [11], whereas the benefits to increase other biodiversity features such as plant species richness and density may require longer periods (<30 years) [49–51]. Therefore, selecting a timeframe that can reflect positive outcomes not only for pollination services is important to foster more sustainable practices. Accounting for the difference on the benefits from restoration, we used a 40 years' time horizon to calculate the net present value (NPV) to determine coffee profitability. This timeframe was based on past studies of rainforest recovery rates from the tropics (i.e., Puerto Rico, Brazil, and Costa Rica, [49–52]). These studies showed that plant species richness increased in abandoned or restored areas [49,50,52] and in some cases aboveground biomass recovering showed similar values to old grown forest [51].

We calculated the NPV of coffee revenue as follows:

$$NPV_i = \sum_{t=0}^{T}((yo_{it} * p_{it}) - c_{it})/(1 + r)^t \tag{2}$$

Where $yo_{it}$ represents expected coffee yield in each $i$ cell (estimated with Eq 1). The price of coffee (average price paid from fruit and dry coffee) is represented by $p_{it}$. The production cost is represented by $c_{it}$ and is specific to the coffee's age, taking into account the establishment cost, the maintenance cost (which is composed of yearly maintenance and coffee renovation maintenance), the harvest cost, and the transportation cost. The discount rate is represented by $r$ set at 9% as suggested by Aylward and Porras [53] for Costa Rica.

It is expected that bee abundances will slowly recover after restoration until their abundances increase and therefore the pollination services they provide are equivalent or closely equivalent to numbers observed in mature forest. For example, connecting crops to natural areas with hedgerows have had positive results for wild bee abundance and diversity. For example, after a year of growing hedgerows in farmland, M'Gonigle and colleagues [54] found bee abundance to have increased in size by 8%. Seven years later, they also found that species

richness had increased compared to sites with no hedgerows. With mature hedgerows that are more than 10 years old, bee abundance was significantly higher when there was natural vegetation near crops (at least 100 m), compared to control sites [55].

As such, different bee abundance data were used to calculate the expected yield provided by pollinators from these restored and intact forest patches. For "intact forest," we used data from pollinator surveys conducted in Costa Rican coffee landscapes reported on in the published literature [28,29,48,56–58]. To estimate bee diversity in "restored forest," we assumed a direct relationship between plant richness and bee abundance, as reported by Kremen and colleagues [59] after a restoration action in Central Valley of California. For this, we used the plant richness reported by Aide and colleagues [60] and bee abundance as reported by Brosi and colleagues [61] in Costa Rica (Table S2.2 and S2.3 in S1 Appendix). Therefore, bee abundances (Table S2.3 in S1 Appendix) and pollination services increased gradually, as this depends on the distance from forest and the time since restoration commenced (S1 Appendix—2.2). We also assumed that bee richness, abundance, and pollination services would be completely restored after 40 years. The second and third components are environmental and physiological factors that determine coffee yield productivity. One of the factors considered in calculating yields was land suitability, where we conducted a suitability analysis of the study region to identify the best areas to grow coffee (taken from López-Cubillos and colleagues [17], S1 Appendix—2.3 shows more information). Finally, age of the coffee is another physiological important factor to be considered when calculating yields, as coffee starts producing fruit from the second year of growth, reaching a peak between years 5 and 6 (see S1 Appendix—2.2). Finally, we considered not only coffee production cost for the profit assessment, but also we calculated the cost of active restoration over the same timeframe, which included establishment and maintenance cost (S1 Appendix—2.2).

## Planning goals and optimization analysis

We used integer linear programming to find the optimal arrangement of land for coffee production, forest, and restoration sites to increase pollination services and maximise forest protection. We explored how to maximise the profits, intact forest retention (ha), and forest restoration (ha) when pollination services were considered. The general form of the optimization is:

$$Maximize: \quad \sum_{k=1}^{K} \sum_{i=1}^{I} NPV_{ik} x_{ik} + \lambda F_{ik} x_{ik} \tag{3}$$

$$Subject\ to: \quad \sum_{j \in M_i} (x_{j1} - m x_{ik}) \geq 0, \forall_i, k \in \{3 - 5, 8 - 10\} \tag{4}$$

$$\sum_{i \in Y} \sum_{k=1}^{K} x_{ik} \geq C, \forall_i, k \in \{3 - 5, 8 - 10\}$$

$$\sum_{i \in R} \sum_{k=1}^{K} x_{ik} \geq F, \forall_i, k \in \{7\}$$

$$\sum_{i \in Z} \sum_{k=1}^{K} x_{ik} \geq P, \forall_i, k \in \{1\}$$

$$\sum_{i \in Y} \sum_{k=1}^{K} x_{ik} \geq N_c$$

$$\sum_{k=1}^{K} x_{ik} = 1$$

$$x_{ik} \in \{0, 1\}$$

Here, we aimed to maximise the NPV and forest restoration and/or protection (Eq 2). $x_i$ in Eq 3 is the decision variable that represents if cell $i$ should be converted either to restoration, to crop expansion or kept as intact forest. $NPV_{ik}$ represents the NPV from agriculture in cell $i$ in zone $k$. Here, $k$ can be any of the following 10 zones: 1. Forest; 2. other land use (in this Costa Rican landscapes is usually pastures or sugar cane); 3. coffee with "high" pollination services as it is adjacent to the foraging distance category 1 from intact forest; 4. coffee with "mid" pollination services as it is falls into foraging distance category 2 from intact forest; 5. coffee with "low" pollination services as it is falls into foraging distance category 3 from intact forest; 6. coffee that do not receive pollination services because it exceeds the bees' foraging distances; 7. restored forest. Zones 8, 9, and 10 represent coffee with "high," "mid," and "low" pollination services (respectively) based on their distance to restored forest (rather than intact forest). $F_{ik}$ represents the amount of forest restored or protected in each cell. $\lambda$ is a factor to scale the importance of forest relative to the NPV, from which we can identify the 3 different strategies ("profit focus," "balanced," and "conservation focus").

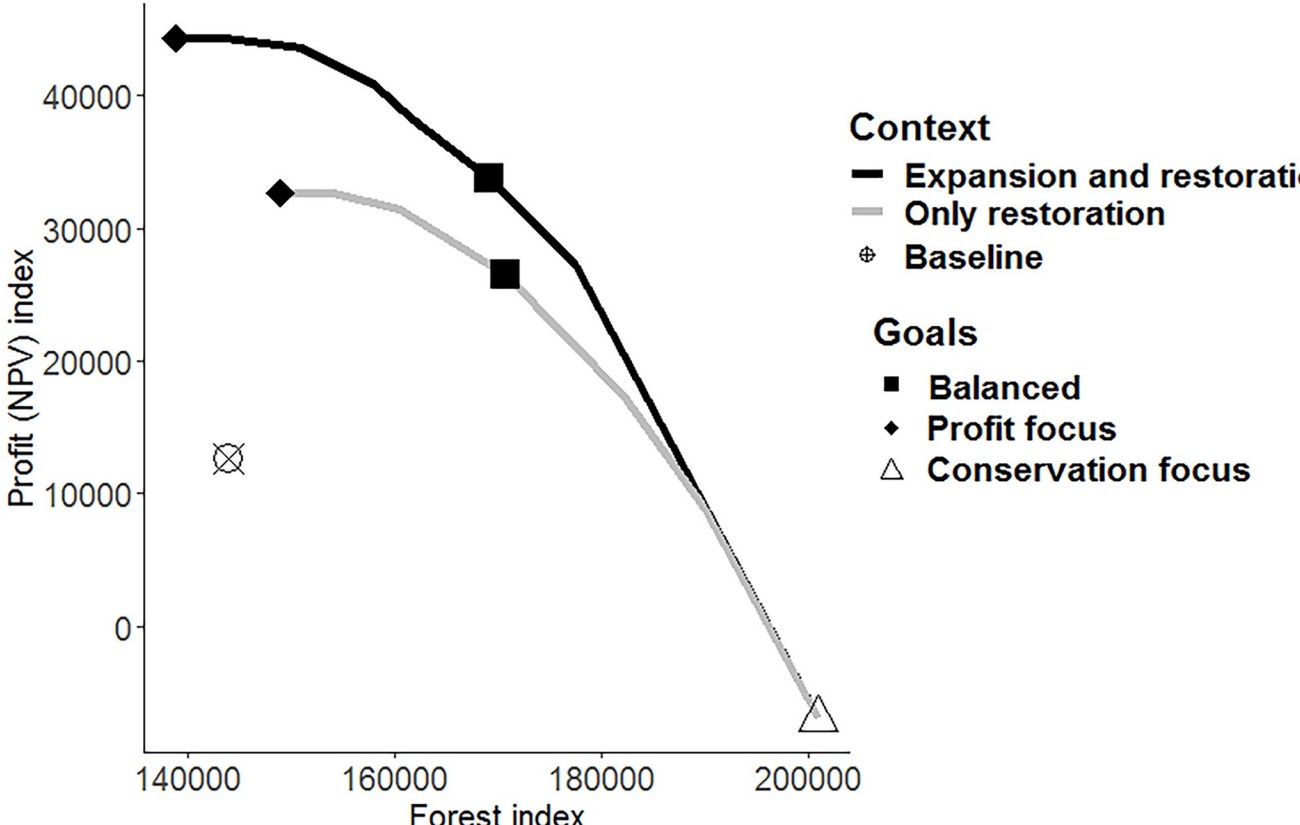

**Fig 3. Trade-offs between profit (NPV) and forest for the end of the restoration horizon (40 years).** Pareto curve for the "baseline," "only restoration," and "expansion and restoration" agricultural contexts. The black and grey lines represent all the scaling values used ($\lambda$). "Profit focus," "balanced," and "conservation focus" goals are represented by diamond, squares, and triangle, respectively. The index is the result of the profit (NPV) or forest for all "planning cells" over the 40-year period. Here, the forest index includes both restored and remanent forest. The data underlying this figure can be found in S2 Data.

The first constraint (Eq 4) ensures that a cell $i$ can be allocated to zones 3 to 5 and 8 to 10 (coffee zones that benefit from pollination) if at least 1 of the proximal cells ($Mi$) are allocated to forest (zone 1 which is $x_{j1}$) and here we set $m$ to 1. $Mi$ refers to all cells with a given radius of cell $i$, and m can take any value between 1 and $Mi$. These radiuses were set at 0, 55, and 663 m from zone 3 to zone 5 and from zone 8 to zone 10. This means that a minimum threshold of 1 cell of forest are proximal to cells zoned for coffee production that receives pollination benefits. The second constraint ensures that the set of cells that have already been converted from forest to coffee (set $Y$ of length $C$) remain as coffee, but are permitted to be allocated to any of the coffee zones (zone 3 to zone 6 and zone 8 to zone 10). This means that once forest is converted to coffee it cannot be converted back. The third constraint ensures that the set of cells that have already been converted from coffee to forest (set $R$ of length $F$) remain as restored forest. This means that once forest is restored it cannot be converted back to coffee. Together, the second and third constraints ensure realism by disallowing the same cell to be converted back and forth between coffee and forest at each time step. The fourth constraint ensures that intact forest (set $Z$ of length $F$) remains forested. Analyses were performed in R version 3.3.1 [62] and solved using the software Gurobi [63].

## Sensitivity analysis

To explore the NPV in a time horizon of 40 years, we varied the discount rate to ±3% from original (9%) as suggested by Aylward and Porras [53] for Costa Rica. As the economic costs and benefits of pollination services are variable [64], we also varied the NPV of coffee and restoration cost by ±25%. Finally, because restoration projects are rarely 100% successful, we used 3 probabilities of restoration failure based on values reported for tropical restoration projects [65–67]: 10%, 30%, and 50%.

## Results

While there are evident trade-offs between economic returns and the total amount of forest in the landscape, both objectives can be increased from the baseline when there is a strategic allocation of restored land (Fig 3). The "conservation focus" goal maximised forest cover, leading to extensive restoration in both contexts. On the other hand, the "profit focus" goal showed higher agricultural profit (NPV) in the "expansion and restoration" context, but with a smaller total forest area than in "restoration only" context.

Comparing the final time step (year 40) with the baseline context, the "conservation focused" goal preserved all intact forest (Fig 2) and increased restored forest by 40% for both "crop expansion and restoration" and "only restoration" contexts (Fig 4D). However, this result has an evident negative effect on profits, with NPV being negative and lower than the baseline (Fig 4C) as restoration completely overtakes the former coffee cropping areas (Fig 2). The "balanced" goal achieved substantial conservation and economic benefits. In this case, restored forest increased approximately 20% for both contexts compared to the baseline (both in the short and long term, Fig 4B and 4D) and includes a mix of large and small restoration patches (Fig 2). Nevertheless, intact forest in the "balanced" goal for the "crop expansion and restoration" context led to a decrease of around 25% of the intact forest in the long term (after 40 years of restoration, Fig 4D) but not in the short term (after 5 years of restoration, Fig 4B). Overall, profits increased approximately 100% or more in both contexts for the balanced goal both in the short and long term (Fig 4A and 4C), with a higher NPV in the "expansion and restoration" context in most of the cases because of the conversion of some intact forest to coffee production. "Profit focused" was clearly the most profitable goal in the long term (after 40 years of restoration, Fig 4C), but it had the poorest outcomes for forest, with intact forest decreasing by approximately

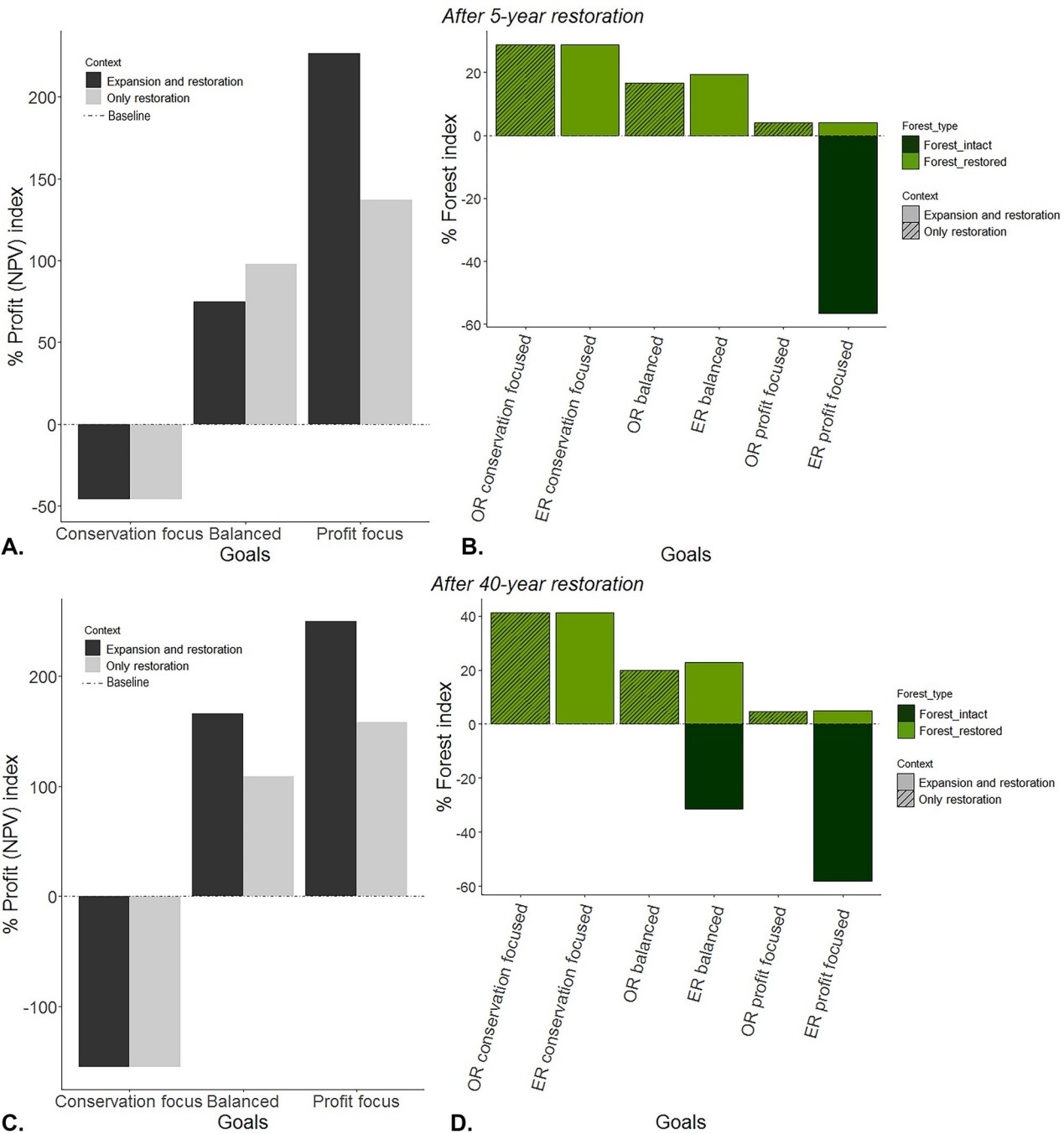

**Fig 4.** The difference in the profit (NPV) index (A and C) and the forest index (B and D) for each goal per context compared to the baseline at the end of the restoration time horizon for 5-year and 40-year restoration outcomes. The dashed line at zero represents the baseline. The data underlying this figure can be found in S3 Data.

60% for the "crop expansion and restoration" context (Fig 4D). However, in a short term (after 5 years of restoration, Fig 4A), a balanced goal in the "only restoration" context turns to be more profitable than the "expansion and restoration" context (98% versus 74%, respectively). It is important to note that even when profit was the only goal, some forest was still restored (approximately 5% in both contexts both in the short and long term, Fig 4B and 4C), but this was primarily in smaller patches relative to the other goals (Fig 2).

Across the time horizon, "total forest" (restored and remanent) increased in all cases relative to the baseline (Fig 5A and 5C), with except when profit alone was prioritised in the "expansion and restoration" agricultural context (diamonds in Fig 5C). This exception is the product of crop expansion into intact forest; however, the forest index progressively increased over time as small patches of coffee are restored to forest (although the baseline level of total forest was still not reached in our 40-year horizon). Most goals generated profits higher than baseline, except when forest restoration was the sole priority (i.e., "conservation focus" goal)

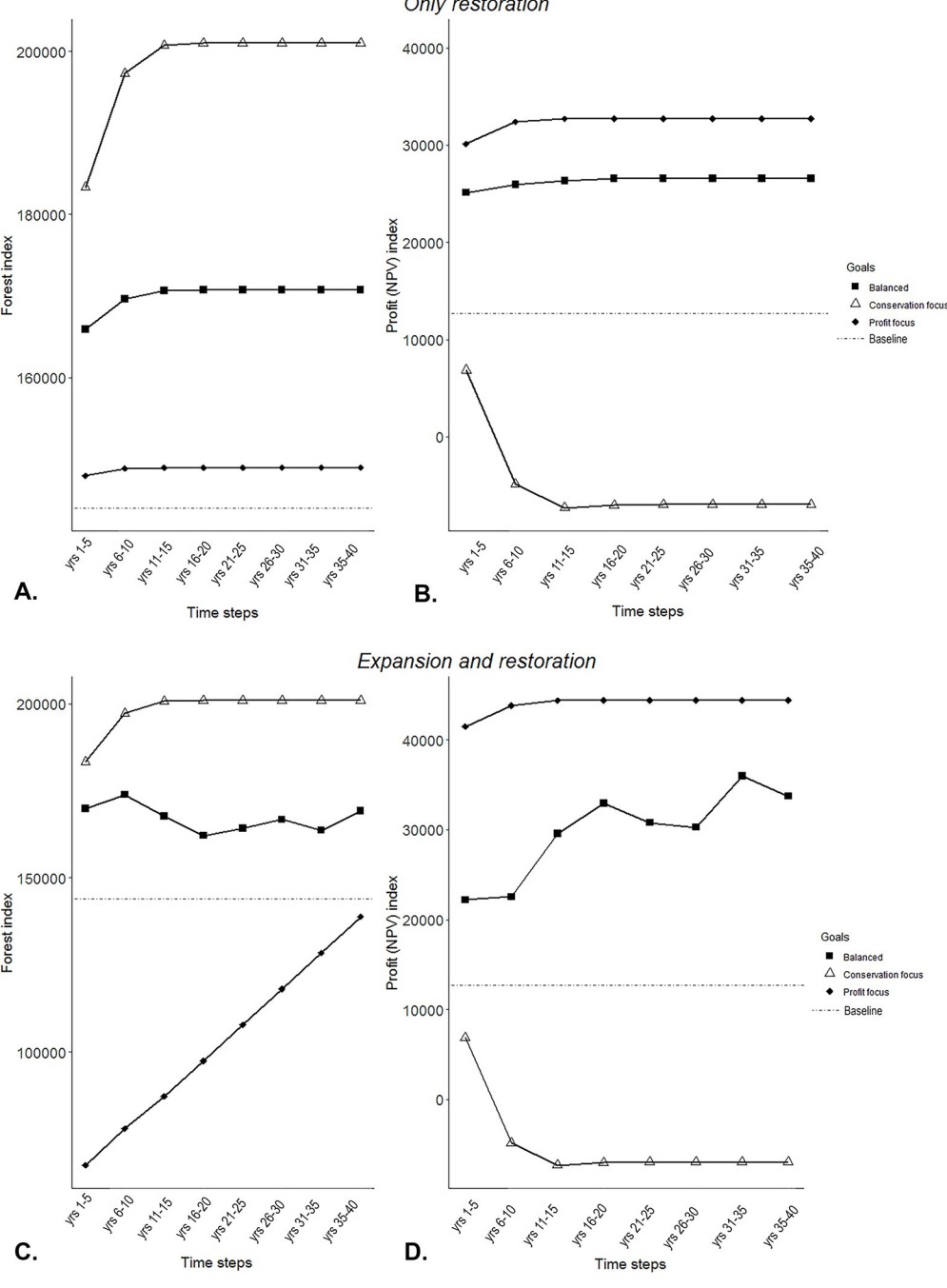

**Fig 5.** Cumulative forest (A and C) and profit (NPV) index (B and D) over the time horizon (40 years) for the only restoration and expansion and restoration contexts. The data underlying this figure can be found in S4 Data.

(Fig 5B and 5D, triangles). A plateau for both forest benefits and profit (NPV) is reached after 20 years for most of the goals (Fig 5 diamonds, triangles, and some of the square symbols). It is important to note that the balanced goal from the "expansion and restoration" context shows the most dynamic pattern for both forest and profit, as it does not reach a plateau over time.

## Sensitivity analysis

As both profits and forest area can fluctuate when key input parameters vary, we performed a series of sensitivity analyses (see Fig 1, "sensitivity box," and Table S2.4 in S1 Appendix). The most sensitive variables (i.e., those that showed the highest and lowest outcomes in Fig 6) were

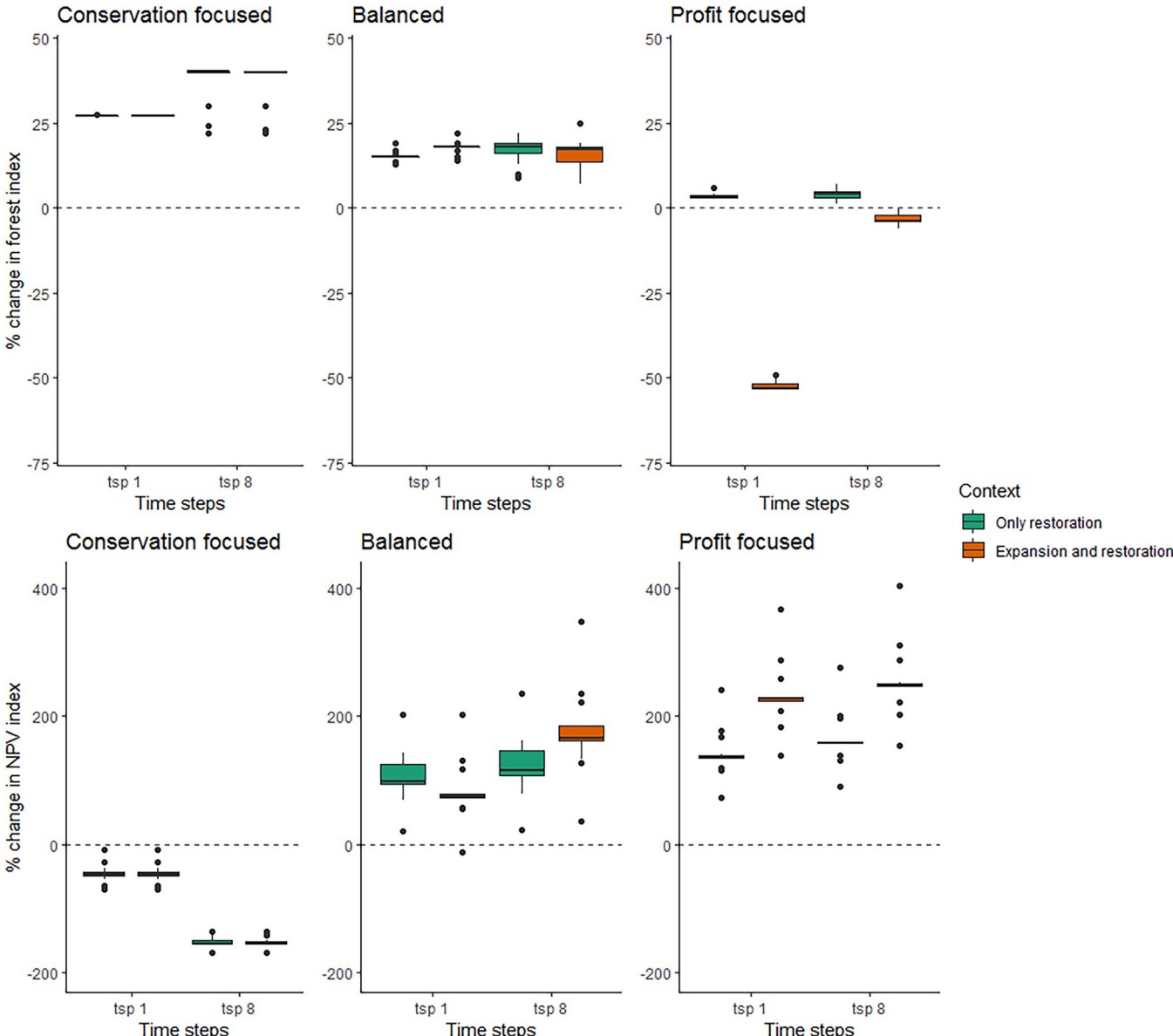

**Fig 6. Changes in forest and NPV among the agricultural contexts based on the sensitivity analyses (see Methods section).** The dashed line indicates the baseline, and the box and whisker plots indicate the variation from all sensitivity analyses. The solid black line represents the median and the lower and upper hinges correspond to the first and third quartiles (the 25th and 75th percentiles). The data underlying this figure can be found in S5 Data.

the "discount rate" and the level of "restoration failure" (Supporting information, Table S2.4 in S1 Appendix). A high discount rate (i.e., 12%) reduces the contribution of economic benefits (NPV) and costs occurring in future periods relative to a lower discount rate (i.e., 6%). On the other hand, having a 50% probability of restoration failure showed the poorest forest outcomes for both "conservation focused" and "balanced" goals (Fig 6, time step 8). Despite the sensitivity of some variables, we can observe a similar trend as the original analysis, where the conservation focused goal provides good outcomes for restoration but poor for profits, and on the contrary, the profit focused goal increases profits but with poor conservation outcomes.

## Discussion

We found that strategic planning can capture the benefits of restoration to agricultural production through pollination services, which can lead to substantial economic benefits: up to a doubling of agricultural profits (NPV), even accounting for the lost production on restored lands. This is contrary to previous work that assessed the economic benefits 50 years post initial restoration that showed that while nonmarket ecosystem services (e.g., cultural, aesthetic, and recreational value) were improved, economic benefits were not achieved in marketed ecosystem services (e.g., carbon markets) [68]. However, this previous work did not consider the ecosystem services to agriculture, such as pollination, which we have shown are crucial in delivering economic benefits. Our results showed an increase in both forest cover and profit for most of the goals (i.e., "balanced" and "profit focused") relative to baseline levels. We have even demonstrated improvements for profit and forest cover in the balanced goal when we allow coffee expansion and restoration at the same time. This highlights that strategically prioritising restoration not only provides biodiversity benefits but also enhances local livelihoods in the tropics [69].

The spatial arrangement of our "balanced" and "profit-focused" goals contained both large contiguous areas of forest and small forest patches within coffee. This suggests a mix of land sharing (many small fragmented forest patches) and land sparing (large areas set aside) is optimal in these landscapes, which is in line with other studies for the conservation of remanent forest in coffee landscapes [17]. Our results appear to differ from previous studies that showed that sparing is better in agricultural landscapes [70]. A result that may be explained by our focus on ecosystem service flows rather than a species focus (e.g., bird and plant biodiversity). Furthermore, pollination services operate over small spatial scales [71] and they also depend on animal movement from forest fragments, the interspersion of remnant or restored forest within croplands is akin to a land sharing approach [72]. However, our findings are in line with emerging literature showing that mixed sharing/sparing landscapes are optimal when we capture the spatial heterogeneity of landscapes, with a mix of remanent forest, restored areas, and agricultural lands [73].

In development frontiers, where further crop expansion may be unavoidable, positive results for biodiversity can still be achieved, despite the loss of some remnant forest. The "balanced" goal increased the total amount of the forest index by 15% and increased agricultural profit by 109% compared to the baseline. However, this is far from a green light for unchecked agricultural development, as our results were only achieved by strategically allocating new coffee production across landscapes, while retaining substantial forest patches. Caution is thus needed with crop expansion, as removing nature to increase economic gains will reduce the resiliency of the ecosystem and increase environmental variability (e.g., longer droughts) and potentially lead to ecosystem collapses if not managed in line with our understanding of the ecology of fragmented landscapes [74]. Furthermore, some landscapes with dense forest cover (≥75% canopy) that are located nearby coffee crops (approximately 1 km) have higher

pollination richness and fruit set due to pollination services [24]; therefore, focusing only on restoration rather than deforestation and restoration at the same time would be the best strategy. Our results show the potential for how restoring lands in agricultural landscapes can achieve environmental conservation outcomes without being an economic burden in the long term.

Accurately calculating the economic benefits from pollination services can be challenging as costs and benefits flow at different times, and multiple key variables can affect the outcome [64]. In this case study, our results were robust to variation in these key variables—positive outcomes for agricultural profits and net forest area relative to the baseline were achieved in most cases. Varying the discount rate used to calculate NPV had the biggest impact on results and as such, it is a factor that must be contemplated carefully when using these rates to plan restoration activities over time [75,76]. Specifically, higher discount rates reduce the contribution of future economic benefits and costs to the NPV calculation and are therefore likely to favour agricultural production (with immediate benefits) over restoration (with a slower accrual of benefits). Alternatively, reducing the cost of restoration by 25% increases the area of forest restored by up to 6% for the "profit focused" goal when only considering restoration. Such cost reductions could be achieved by using a mix of passive [77] and active restoration approaches. However, as benefits may be slower to accumulate with passive restoration, this remains an important area for further research.

Our model assumes that bee abundance will fully recover after 40 years in tropical forest and that the pollination benefit from restoration will be achieved between 6 to 15 years of being restored, as substantial pollination benefits are provided by common species, such as *Apis mellifera* [29]. These timelines are estimates for our case study system and should not be used as general assumptions that can be applied to other systems. The literature about the benefits of restoration on bee abundance is mixed, with some studies aligning with our assumptions and others unaligned. It is much better to set these timelines according to the known understanding of each landscape and forest system to which our method is applied and even then, care should be taken to acknowledge that recovery times can and do vary extensively within and across systems. Positive results for bees' population recovery after small-scale restoration approaches (such as hedgerow restoration) have been found in intensive agricultural landscapes after 7 to 10 years of restoration [54,55,78]. In restored sections of riparian forest, however, Gutiérrez-Chacón and colleagues [79] and Williams and colleagues [80] found that bee communities are not similar to reference riparian areas or forests. Gutiérrez-Chacón and colleagues [79] highlighted that species composition showed progress in recovery in this system and suggested more time (>13 years) would be needed in addition to better management actions, such as fencing to protect habitat from cattle, to see a return to a species composition similar to reference sites.

We found that even when the sole goal of restoration was to maximise profit, it was still optimal to restore forest patches throughout the coffee landscape, solidifying the importance of pollination to production. However, economic benefits from restoration take time to materialise; for example, the "balanced" goal for the "expansion and restoration" context took considerable time to achieve the highest profitability when we allow restoration and forest expansion at the same time. Therefore, understanding these temporal differences in profitability can help to identify leverage points to enhance the uptake of restoration. Payment for ecosystem services (PES) schemes could compensate landholders for the initial restoration actions, which would help address this temporal mismatch in economic costs and benefits [81–83]. Indeed, these types of schemes have been successful in other tropical agricultural lands where farmers receive financial incentives in line with the profits they would have received if they would have continued agricultural production (i.e., opportunity costs [84]).

Our restoration solutions could feasibly involve multiple landholders in restoration efforts, some of which are likely to be smallholders (as is very common in Costa Rica [85]). The inclusion of many small landholders could represent a challenge for implementing a PES scheme, as this can increase transaction costs and reduce the potential for coordinated landscape management. If the transaction cost exceeds, or is close to, the PES incentive payment, the project may not be viable [86]. In such cases, farmer collectives may form an avenue to reduce transaction costs and maintain the benefits of these payment schemes [81,85]. In addition, more extensive and ongoing PES may be required to incentivise the restoration of the larger forest patches—meaning greater conservation benefits as seen in our "conservation" and "balanced" goals—as they show larger trade-offs with profitability.

Strategic planning for restoration is vital, especially in the global south, which contains many biodiversity hotspots alongside important regions for agricultural development and community livelihoods [82,87]. Therefore, restoration should be planned in a manner that minimises the impacts on food production and where farmers can benefit from ecosystem services [88]. Unfortunately, despite the importance of pollination services, their inclusion in land-use plans and policymaking is usually ignored or poorly implemented [89,90]. Our spatial framework is a step towards reconciling the often-divergent objectives of nature restoration and agricultural production by including pollination services. We found that the strategic allocation of restored habitat within croplands can synergistically enhance biodiversity and agricultural production, and that some level of restoration is ideal even if solely aiming to maximise profit. This is relevant to many landscapes across the globe with vast swaths of monoculture plantations, where strategic restoration could improve outcomes for both pollinators [91] and landholder profits. Our framework integrates the spatiotemporal dynamics inherent to ecological processes and economic information, which is a key advance to guide decision-making in agricultural landscapes around the world [92]. Landscape restoration for targeted species, like wild bees, shows that we can have potential win-wins for both biodiversity and people in heavily human-modified landscapes.

## Supporting information

**S1 Appendix. Supporting methods and results.**
(DOCX)

**S1 Data.** Supporting data for Fig 2 Tabs 1 to 6: i. Data_Fig_2 Baseline, ii. Data_Fig_2 Conservation focused, iii. Data_Fig_2 Expan and resto–balanced, iv. Data_Fig_2 Expan and resto–profit, v. Data_Fig_2 Only restoration–balanced, and vi. Data_Fig_2 Only restoration–profit.
(XLSX)

**S2 Data. Supporting data for Fig 3.**
(XLSX)

**S3 Data. Supporting data for Fig 4.**
(XLSX)

**S4 Data. Supporting data for Fig 5.**
(XLSX)

**S5 Data. Supporting data for Fig 6.**
(XLSX)

**S6 Data.** Supporting data for Fig 2.1 in S1 Appendix.
(XLSX)

## Acknowledgments

We are thankful with Dr. Rebecca Chaplin-Kramer and Dr. Taylor Ricketts for making comments on early versions of the manuscript.

## Author Contributions

**Conceptualization:** Sofía López-Cubillos, Eve McDonald-Madden, Margaret M. Mayfield, Rebecca K. Runting.

**Formal analysis:** Sofía López-Cubillos, Rebecca K. Runting.

**Investigation:** Sofía López-Cubillos, Eve McDonald-Madden, Rebecca K. Runting.

**Methodology:** Sofía López-Cubillos, Eve McDonald-Madden, Margaret M. Mayfield, Rebecca K. Runting.

**Project administration:** Sofía López-Cubillos.

**Resources:** Sofía López-Cubillos.

**Supervision:** Eve McDonald-Madden, Rebecca K. Runting.

**Writing – original draft:** Sofía López-Cubillos, Eve McDonald-Madden, Margaret M. Mayfield, Rebecca K. Runting.

**Writing – review & editing:** Sofía López-Cubillos, Eve McDonald-Madden, Margaret M. Mayfield, Rebecca K. Runting.

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
