## [Editor Report · Decision Letter 0]

22 Sep 2022

Dear Dr López-Cubillos, 

Thank you for submitting your manuscript entitled "Optimal restoration for pollination services increases forest cover while doubling agricultural profits" for consideration as a Research Article by PLOS Biology.

Your manuscript has now been evaluated by the PLOS Biology editorial staff, as well as by an academic editor with relevant expertise, and I'm writing to let you know that we would like to send your submission out for external peer review.

Once your full submission is complete, your paper will undergo a series of checks in preparation for peer review. After your manuscript has passed the checks it will be sent out for review. To provide the metadata for your submission, please Login to Editorial Manager (https://www.editorialmanager.com/pbiology) within two working days, i.e. by Sep 26 2022 11:59PM.

Kind regards,

Roli Roberts

Roland Roberts, PhD

Senior Editor

PLOS Biology

rroberts@plos.org

---

## [Decision Letter · Decision Letter 1]

14 Nov 2022

Dear Dr López-Cubillos,

Thank you for your patience while your manuscript "Optimal restoration for pollination services increases forest cover while doubling agricultural profits" went through peer-review at PLOS Biology. Your manuscript has now been evaluated by the PLOS Biology editors, an Academic Editor with relevant expertise, and by two independent reviewers.

In light of the reviews, which you will find at the end of this email, we are pleased to offer you the opportunity to address the comments from the reviewers in a revision that we anticipate should not take you very long. We will then assess your revised manuscript and your response to the reviewers' comments with our Academic Editor aiming to avoid further rounds of peer-review, although might need to consult with the reviewers, depending on the nature of the revisions.

IMPORTANT:

a) Please attend to the requests from the reviewers. Note that the Academic Editor has kindly provided some extra comments (see the foot of this email) which you might find helpful in guiding your revisions.

b) Please comply with our Data Policy; specifically, we need you to supply the numerical values underlying Figs 2, 3, 4AB, 5AB, 6, S2.1, either as a supplementary data file or as a permanent DOI’d deposition.

c) Please cite the location of the data clearly in all relevant main and supplementary Figure legends, e.g. “The data underlying this Figure can be found in S1 Data” or “The data underlying this Figure can be found in https://doi.org/XXXX”

We expect to receive your revised manuscript within 2 months. Please email us (plosbiology@plos.org) if you have any questions or concerns, or would like to request an extension. 

**IMPORTANT - SUBMITTING YOUR REVISION**

*Resubmission Checklist*

*Published Peer Review*

*PLOS Data Policy*

*Blot and Gel Data Policy*

Sincerely,

Roli Roberts

Roland Roberts, PhD

Senior Editor

PLOS Biology

rroberts@plos.org

REVIEWERS' COMMENTS:

Reviewer #1: 

There is a lot to like in the paper. In brief the authors present a framework to identify the best places for restoration in coffee plantations over a 40-year period following restoration. They find that their restoration treatment increases forest area by 18% while doubling farmer profits after 40 years. It's an interesting question to ask, using an interesting approach too, especially with regard to the spatial arrangement of the forest patches. As a non-modelling field ecologist most of the parameters that went into the models made intuitive sense. I have a couple of major issues (which may just be the MS not being clear enough for the non specialist reader) and a few minor issues. 

MAJOR ISSUES

Line 117 states "It is expected that bee abundances will slowly recover after restoration until they are equivalent or closely equivalent to numbers observed in mature forest". I'm not convinced that bee numbers will ever restore to that present in a mature forest - as it's not clear at this point how big these restored area are? Forest area will likely affect how many species they have, as will how close patches of forest are and potentially their shape too (due to edge effects). I'm left wondering if bee abundance in a patchy and fragmented forest (Fig 3) ever be the same as that in mature forest (which by definition sounds a contiguous habitat rather than a patchy one)? It doesn't need to be, it just needs to be considerably better than what was there before, which is a realistic outcome.

Line 185: Is a 40 year timescale at all realistic in a working agricultural habitat? That is an entire working life of a farmer and I can't see any enthusiasm from farmers for a scheme with a payback period of 40 years? Or a government incentive that would work over this time period either? While I can see that this is based on the time taken for tropical forests to recover, there are very likely to be positive changes on biodiversity well before the 40 years are up? On line 235, it does say something about 20 years, but see my comments on this section in my next point. In the discussion (line 264) the authors talk about predictions 50 years after restoration work by other authors, so maybe this timescale is the norm in this field, in which case the reader needs a bit more information on the timescales in the introduction. Otherwise, the paper risks reading like an interesting theoretical exercise, but not a solution that is of any practical use, as farmers simply can't wait 40-50 yrs for a return on investment (here land lost to production). Nor does it have much real relevance to solving the biodiversity crisis if we have to wait 40 years for an outcome..

Line 235: Here it says that a plateaux for forest benefits and profit are reached after 20 years, but I had a hard time understanding "Figure 5, diamonds, triangles and some of the square symbols". Is it possible to calculate a break-even point as done (using a much simpler approach) in Blaauw et al (ref 8 in MS)? Or is this what figure 5 shows, and I just can't make it out?

MINOR ISSUES

Line 50 The authors say that it's not clear if restoring the native flora next to blueberries is cost effective citing Benayas & Bullock 2012 - however Blaauw & Isaacs 2014 clearly shown that it is cost effective for blueberries and the restoration work can pay for itself remarkably quickly. The authors point this out on line 60, which looks like contradiction to the Benayas & Bullock refference. Similarly, Pyewell et al. have shown that taking land out of production can quickly become cost effective eg. Pyewell et al. 2016 http://dx.doi.org/10.1098/rspb.2015.1740

Line 126: Why is coffee bush age important to bees?

Line 140 "To solve this problem" it's not clear what problem is being solved?

Reviewer #2:

In this study, Sofía López-Cubillos et al. investigated the benefits of restoration to coffee production through pollination services. More specifically, the authors modelled the trade-offs between the economic returns of coffee farming and the total amount of forest in a coffee agriculture landscape in Costa Rica over 40 years. The study found that economic return from coffee production as well as the amount of forest in the wider landscape could be increased with strategic planning of the landscape management activities. 

The manuscript is well written, and the spatial optimisation framework offers interesting insights into a highly topical issue: the role of agriculture in forest landscape restoration. The paper is well structured, the figures are adequate, and the presentation is clear. I find the findings interesting and overall, this is a relevant work that demonstrates that restoration does not necessarily have to be in competition with agricultural crop production. There are, however, several improvements that can be made, largely to enhance the overall rationale and ensure that the findings are fully supported by the methodology.

General comments:

Introduction

The overall rationale of the introduction lacks detail and could benefit from a more comprehensive review of up-to-date literature. To strengthen the introduction, I recommend the authors expand more on the definition of restoration in the context of this study and the role of natural habitats in increasing crop pollination. I missed reference to the global assessment on the value of biotic pollination and dense forest for fruit set of Arabica coffee by Moreaux et al. 2022 and a more detailed explanation of the role of wild pollinators in coffee production. Since there has been a lot of research on the dependency of coffee crops on pollinators, I would suggest including a more detailed overview of the current state of knowledge. Also, authors could include more detail about the context and different types of restoration, e.g. in L57f, e.g. a sentence about natural regeneration and active restoration (tree planting) and justify why the focus here is on active restoration.

Methods

The spatial optimisation framework introduces an original way of assessing the benefits of strategic land management to crop production and forest conservation. However, the methodology needs to be clarified in several instances. For example, could the authors justify more clearly why land suitability, pollination services and coffee bushes' age were included as the factors to estimate coffee yield, and how the factors were weighed? Also, I think the differences between the "agricultural context" and "goals" presented in Figure 1 need to be more clearly explained. For example, what is the difference between the "only restoration" agricultural context and the "conservation focused" goal? Further, could the "agricultural context" be understood as different scenarios (e.g. similar to De Pinto et al. 2020, a global analysis of the effects of involving crop production in restoration efforts, who based their modelling approach on a comparison of a business-as-usual scenario with a series of alternative scenarios)? I would urge the authors to clarify their approach in the methods section.

Results and discussion

On several occasions, the authors make statements about their findings that are not fully aligned with their research questions and method. I recommend the authors carefully revise the results and discussion to ensure that the reported findings are fully supported by the methodology. For example, L267f "[…] ecosystem services to agriculture, such as pollination, which we have shown are crucial in delivering economic benefits". I suggest the authors rephrase this statement, as their results do not show the importance of pollination in delivering economic benefits, but rather use this as a critical assumption for the spatial optimisation framework. In the abstract, authors state that strategic restoration can increase forest cover by 18%, however this result is not mentioned in the results section (only in Table S2.4 under % total forest change compared to baseline). Is it possible to report this in the results section as well?

Specific comments: 

* L26f: Rephrase this slightly, e.g. 75% of the global leading food crops, and add the reference in the main text (Klein, A., B. Vaissière, J. Cane, I. Steffan-Dewenter, S. Cunningham, C. Kremen, and T.Tscharntke. 2007. Importance of pollinators in changing landscapes for worldcrops.)

* L31: from restored vegetation or generally from vegetation? Maybe the authors could put restored in brackets?

* L50 / L60f and L70f: Is the study by Blaauw and Isaacs (2014) the only one that looks at benefits of restoration to biodiversity and crop productivity? Would it be possible to expand a bit more on this and include other literature?

* L32: determine 

* L79: 3) How important is agricultural expansion to these outcomes? (Capital H)

* Examples of restoration benefits to crop yield: mention blueberries

* L107f: rephrase slightly, for example "pollination services, land suitability and coffee bushes' age were selected as the factors to estimate coffee yield" Why only these ones, justify

* L281: Although PES are a very interesting solution for the mismatch in temporal economic costs and benefits of restoration, it is not central to the findings of this study. Shorten the section or include it later on in the discussion, after the results of this study have been discussed in more detail and embedded in wider literature

* L283: typo: types OF schemes

* L300: not full sentence, rephrase

* L342: some "level" of restoration ?

* L531: not full sentence, rephrase

* L535: typo: Costa Rice (not Corsta Rica)

COMMENTS FROM THE ACADEMIC EDITOR:

Both reviewers are experts in the field and accordingly, their suggestions will be extremely useful for the authors to improve the reach and potential of their MS for PLoS Biol readership. The authors should consider addressing their queries point-by-point in a thoroughly revised version.

Reviewer #1 has a couple of queries that concern with the parametrization of the models and the threshold values of species abundances after restoration. In addition the point is raised of the temporal span of 40 yrs and its realism in connection with ecological recovery processes. I think these are points that can be carefully addressed in the revised version.

Reviewer #2 points out general improvements that can be easily added. For instance, a better updated literature review regarding recent papers missing that can broaden the interest of the MS for PLoS Biol readers. In addition there are concerns regarding the need of more detailed methods descriptions, eg, factors included to estimate coffee yield and the weights given. Also, improving the definition of "agricultural context" in relation to model estimation.

An additional relevant point, regarding the results presentation is that the authors need to link more closely their inferences and conclusions to the actual data and analysis reported. I agree with rev #2 that there is some disconnection now. This can be easily addressed and will result in a better packaged study.

---

## [Decision Letter · Decision Letter 2]

17 Mar 2023

Dear Dr López-Cubillos,

Thank you for your patience while we considered your revised manuscript "Optimal restoration for pollination services increases forest cover while doubling agricultural profits" for publication as a Research Article at PLOS Biology. This revised version of your manuscript has been evaluated by the PLOS Biology editors, the Academic Editor and one of the original reviewers.

Based on the review and our Academic Editor's assessment of your revision, we are likely to accept this manuscript for publication, provided you satisfactorily address the remaining points raised by the reviewer and the following data and other policy-related requests:

a) Please attend to the remaining requests from reviewer #2.

b) Many thanks for providing the underlying data in “Supplementary_mat_Plos_Biol_2023.xlsx” - please could you re-name this file "S2_Data" and change the citations in the Figure legends from "The data underlying this Figure can be found in Appendix S2" to "The data underlying this Figure can be found in S2 Data"?

We expect to receive your revised manuscript within two weeks. 

*Published Peer Review History*

*Press*

Sincerely,

Roli Roberts

Roland Roberts, PhD

Senior Editor,

rroberts@plos.org,

PLOS Biology

DATA NOT SHOWN?

REVIEWER'S COMMENTS:

Reviewer #2:

The authors have addressed most of my comments, and the introduction and methods are much clearer now, supporting the relevant and interesting findings of this study. However, there are a couple of minor points that need to be addressed before the manuscript is ready for publication:

Abstract: 

* L33f: The authors have clarified the methods and results, which are now better aligned. However, the authors write in the abstract: "We present a novel planning framework to determine the best locations for restoration in agricultural landscapes while accounting for yield improvements over 40-years following restoration." From my understanding of the methods and results, this does not align fully with what is described in the manuscript, as the study did not aim to determine where the best locations are, but rather determine the best strategic allocation of restoration in agricultural landscapes (spatial arrangement rather than location). I would suggest the authors clarify this in the abstract.

Introduction:

* L51f: In response to my previous comment to reference the global assessment on the value of biotic pollination and dense forest for fruit set of Arabia coffee by Moreaux et al. 2022, the authors have referred to the work of Moreaux et al. 2022 in the added sentence "Indeed, while coffee can self-pollinate, the yields of Coffee arabica in particular can increase from 10-30% due to bee pollination (24)". However, this statement is not a finding of Moreaux et al.'s work, but a citation of other publications in Moreaux et al.'s article, see "Coffee is able to self-pollinate, but it significantly benefits from biotic pollination with reported fruit set increases of 10-30% in Coffea arabica L., as compared to self-pollination (Hipólito et al., 2018, Klein et al., 2003a, Saturni et al., 2016)." I recommend the authors clarify this and cite the original articles. Also, I suggest the authors have a closer look at the global assessment of the value of biotic pollination and dense forest for fruit set of Arabica coffee, as the findings of that study are very relevant to the rationale of this study.

Minor comment: short title has a typo (restoration)

---

## [Editor Report · Decision Letter 3]

4 Apr 2023

Dear Sofia,

Thank you for the submission of your revised Research Article "Optimal restoration for pollination services increases forest cover while doubling agricultural profits" for publication in PLOS Biology. On behalf of my colleagues and the Academic Editor, Pedro Jordano, I'm pleased to say that we can in principle accept your manuscript for publication, provided you address any remaining formatting and reporting issues. These will be detailed in an email you should receive within 2-3 business days from our colleagues in the journal operations team; no action is required from you until then. Please note that we will not be able to formally accept your manuscript and schedule it for publication until you have completed any requested changes.

Sincerely, 

Roli

Senior Editor

PLOS Biology

rroberts@plos.org